# Selected Approaches to the Assessment of Environmental Noise from Railways in Urban Areas

**DOI:** 10.3390/ijerph18137086

**Published:** 2021-07-02

**Authors:** Miroslav Němec, Tomáš Gergeľ, Miloš Gejdoš, Anna Danihelová, Vojtěch Ondrejka

**Affiliations:** 1Department of Physics, Electrical Engineering and Applied Mechanics, Faculty of Wood Sciences and Technology, Technical University in Zvolen, T.G. Masaryka 24, 96001 Zvolen, Slovakia; 2Forest Research Institute, National Forest Centre, T.G. Masaryka 22, 96001 Zvolen, Slovakia; tomas.gergel@nlcsk.org (T.G.); vojtech.ondrejka@nlcsk.org (V.O.); 3Department of Forest Harvesting, Logistics and Ameliorations, Faculty of Forestry, Technical University in Zvolen, T.G. Masaryka 24, 96001 Zvolen, Slovakia; gejdos@tuzvo.sk; 4Department of Fire Protection, Faculty of Wood Sciences and Technology, Technical University in Zvolen, T.G Masaryka 24, 96001 Zvolen, Slovakia; danihelova@acoustics.sk

**Keywords:** environmental noise, prediction method, rail transport, sound pressure level, urban areas

## Abstract

Rail transport is the second most important way of transporting people and freights by land in the European Union. Rail noise affects around 12 million people in the European Union during the day and around 9 million at night. There are two possible ways to assess environmental noise: noise measurement in situ and prediction using mathematical models. The aim of the work is based on the performed measurements and selected noise predictions to evaluate the accuracy of the prediction models and assess their sensitivity to various aspects. Two measuring points in the Banská Bystrica Self-Governing Region, within Slovakia, were selected for measurement, which is characterized by increased mobility of the population. For prediction, the two methodologies were selected (Schall 03 and Methodical instructions for the calculation of sound pressure level from transport). The results show that the Schall 03 method is sensitive to the measurement location (the value reaches half of the significance level) and to the location–period interaction. The second prediction method is sensitive to systematic error (absolute term) and, such as Schall 03, to the location–period interaction. This method systematically overestimates the results. Results showed greater accuracy of both prediction models compared to the measured noise values than the results of the authors in other countries and conditions.

## 1. Introduction

Noise is one of the basic environmental factors that affect the quality of the living and working environment [1,2,3,4]. Long-term exposure of the population to excessive noise has a proven negative effect on human health, not only on the auditory system but also on other organs [5,6,7,8]. Attention must therefore be paid to reducing the noise burden on the population. Assessing the negative impact of noise on humans is also challenging because the sensitivity (subjective but also objective) of each individual to noise is different. For this reason, generally accepted methodological procedures for its assessment were developed with the selection of suitable parameters and their maximum permissible values [9,10,11,12].

Rail transport is the second most important way of transporting people and freights by land in the European Union. The share of inland rail freight transport in total inland freight transport in the EU in 2018 was around 18.7% (Slovakia 32.6%) [13]. The passenger share from rail transport in the EU in total passenger transport was almost 8% in 2018 (Slovakia 9.9%). According to information from the noise mapping in the Member States collected by the European Environment Agency (EEA) in 2010, rail noise affects around 12 million people in the European Union during the day and around 9 million at night. In reality, these data are undoubtedly higher, as the EEA’s European noise mapping initiative targets only bigger agglomerations and major railway lines with thousands of trains per year. The problem of rail noise is concentrated in agglomerations where the volume of rail transport is highest [14]. In Slovakia, approximately 600 million euros should go to the modernization of railway infrastructure from the European recovery and resilience plan, which should significantly uprate and improve rail transport.

There are two possible ways to assess environmental noise. The first is the realization of noise measurement (“in situ” method). The second option is noise prediction using mathematical modeling. Using these, various prediction models were developed, which are less accurate, but in the planning phase, they are the only way to assess the future noise situation [15,16,17,18,19]. Predictive models tend to be part of the software that models noise in the living or working environment. Based on the prediction results, it is possible to graphically display the noise descriptors in different places of the selected area (noise maps) using special software [20,21,22,23,24].

Under Directive 2002/49/EC relating to the assessment and management of environmental noise, all European countries are obliged to model their environmental noise levels in heavily populated areas. Some countries have their national method, to predict noise but most have not created one yet. The recommendation for countries that do not have their model is to use an interim method [25]. Noise from railway transport was modeled, e.g., [26,27].

Rail transport is generally considered to be more environmentally friendly than car or air transport. From the point of view of noise emissions, it is considered to be the least disturbing. This is given by the frequency range and nature of noise from individual traffic sources [28,29].

The aim of the work was to compare the measurements of sound pressure level A from the railway transport with the results of the selected prediction methods and evaluate their sensitivity to systematic error, the place, and the time of measurements.

## 2. Materials and Methods

### 2.1. Measurement Locations

Two measuring points in the Banská Bystrica Self-Governing Region, within Slovakia, were selected for measurement, which is characterized by increased mobility of the population. One measuring point is located in the central part of Zvolen (marked “ZV”—Figure 1) on a busy railway line connecting the city of Zvolen with the city of Banská Bystrica, near the local hospital. The second measuring point is located in Banská Bystrica near the railway station, in the industrial zone of the city (marked “BB”—Figure 2). This measuring point was chosen so that it was possible to measure and predict noise from two railway lines simultaneously. Measurements and predictions at both measuring points were carried out repeatedly, because since 17 November 2014, based on the decision of the Government of the Slovak Republic, the Railways of the Slovak Republic have been carrying out free transport of selected population groups (students, pensioners). Based on this decision, an increase in the number of persons transported by rail was expected. Therefore, passenger rail transport was strengthened, with the assumption of an increase in noise in the vicinity of railway lines. All measurements were performed during the reference time interval “day” from 6:00 to 18:00 (i.e., 12 h).

In Zvolen, the measurement was performed using a sound pressure analyzer placed on a tripod at a distance of 7.5 m from the railway axis (1.5 m above the terrain) perpendicular to the track axis. The height of the measured point was 3.5 m above the track. Measurements in Zvolen (ZV) took place on 3 October 2011 and 24 April 2015. The results of measurements from this measuring point (ZV) were published in the works of Němec et al. [10,30,31]. The data from these measurements are used in the statistical analysis, therefore the results are repeatedly partially published in this paper.

In Banská Bystrica, the sound pressure analyzer was located at a distance of 7.5 m from the axis of track no. 172, which corresponded to a distance of 15 m from the axis of track no. 170 (Figure 2). The height of the microphone was 1.5 m above the surrounding terrain. The microphone was situated perpendicular to the axis of the track. Measurements in Banská Bystrica (BB) were performed on 20 March 2014 and 14 June 2016. Passenger rail transport dominates on all selected lines.

### 2.2. Measurement Methodology and Prediction Methods

Meteorological and other conditions were suitable to measure on all mentioned days. Each measurement day, time of arrival, time of crossing, type of train, number of wagons, the average speed of movement, and type of locomotive were recorded. The equivalent sound pressure level A (*L*_Aeq,12h_) from the railway transport for the reference time “day” at the given measurement location was calculated according to the ISO 1996-2: 2017 [32]. Subsequently, the equivalent sound pressure level A was determined for each hour (from all train crossings). The measurements were performed using a Brüel & Kjær 2270™ hand-held two-channel sound pressure analyzer (Figure 1). The accessories of the sound analyzer are microphone type 4189 [33].

Most of the known prediction models are based on the calculation of the determinant—*L*_Aeq_ of the equivalent sound pressure level A. This quantity is calculated for a certain so-called reference distance from the noise source (in the case of traffic from the axis of the traffic flow) and subsequently, it is specified for the given place, time, and conditions through corrections. Differences in individual methods are caused by different types of corrections and their application.

Concerning the conditions of railway transport in Slovakia, its intensity, parameters, level of technology, we chose two prediction methods that best correspond to the given conditions and have been used for a long time in Slovakia and the Czech Republic. The method Schall 03 [34] and the method “Methodical instructions for the calculation of sound pressure level from transport” (designation “MPVHD”) [35] were chosen.

The measurement methodology, as well as the calculation of predictions, were performed in the same way as in the published research in the papers Němec et al. [10,31]. The literature [34,35] characterize the exact methodological procedure, which was followed in the measurements performed in this research. The basic comparison of both prediction methods is in Table 1.

*P* is the percentage share of disc brakes, *l* is the total length of all trains passing through a given place for 1 h, *V* and *v* are the train speeds in km/h and *z* is the number of all carriages (locomotives and wagons). All corrections are calculated in dB. The Schall 03 method also contains other corrections. In the conditions of the Slovak Republic, the *D_Fz_*, *D_Tt_* corrections to the type of wagon and the type of track do not apply. *D_Br_* correction considered bridges and their effect on noise (+3 dB). The correction *D**Lc* is + 5 dB in the case of a railway junction and the *D**Ra* correction is a correction for an annoying squeaking sound that occurs when a trainset passes on a track with a small radius of curvature (from 0 dB to 8 dB).

The Schall 03 methodology is used in the prediction of noise pollution from railway transport, in addition to Germany, also in the territory of other states of the European Union (these states also include the Slovak Republic). In the conditions of the Slovak Republic, it is adjusted by introducing time intervals day, evening, and night. We also do not apply the *D**_Tt_* correction in the form of a 5 dB track bonus. If the tracks have on average at least one irregularity per 100 m, then these correction values *D_Fz_* shall be increased by 2 dB. The technical guidance also contains typical data on lengths, train speeds, and the proportion of disc brakes for each category of the train. Other applied national corrections (crossing of tracks, bridges) were not necessary for our research [36].

The determining quantities (*L*_Aeq,1h_), which were obtained by prediction using both methods, were compared with the values obtained by measurements at both measuring points at both times (after individual hours of measurement). Statistica software (Statsoft Inc., version 13, Prague, Czech Republic) was used for statistical testing and analysis.

## 3. Results

A comparison of the determining values obtained by measuring 3 October 2011 with the predictive values of selected models after individual hours for the measuring site Zvolen (ZV1) is shown in Figure 3. For easier orientation and a clearer demonstration of the differences between the individual methods, the equivalent sound pressure levels A were connected in the given intervals. It is clear from Figure 3 that the tendency of the determinants from the measurements and predictions are similar. Both ZV1 and ZV2 measurements were more accurately predicted by Schall 03.

Figure 4 shows a comparison of the values of the determining quantities (*L*_Aeq,1h_) from the measurement performed on 24 April 2015 (ZV2).

The first measurement in Banská Bystrica (BB1) was carried out on 20 March 2014. The meteorological conditions were suitable for the measurement. More detailed information on the composition, number, and types of trains is in Table 2.

Both for the assessed place in Zvolen and for the assessed place in Banská Bystrica, the values of determining quantities (*L*_Aeq,1h_) obtained by predictions were compared with the values obtained by measurement at the measuring point in Banská Bystrica. Figure 5 again presents a graphical evaluation of the results obtained by both prediction methods and measurements in one-hour time intervals on 20 March 2014 (BB1).

The highest noise burden was reached between 12:00 a.m. and 01:00 p.m. During this hour, the highest intensity of rail traffic was recorded. A total of five trains passed through the section, of which two were freight (they had 22 and 24 wagons). The minimum noise exposure was achieved between 11:00 a.m. and 12:00 a.m. when only two trains passed through the section. The equivalent sound pressure level A at the reference time “day” for BB1 was *L*_Aeq,12h_ = 63.27 dB.

The tendency of the determining quantities obtained by the prediction methods and measurements is similar, except for the value in the time from 2:00 p.m. to 3:00 p.m., when a train with a low running speed of 10 km/h crossed the line. The prediction methods could not take sufficient account of this exceptional situation. The prediction methods are based on standard running speeds and cannot take into account, in particular, the starting and braking of trains. Train passing speeds ranged from 10 km/h to 70 km/h (average speed was 45 km/h).

For the measurement location BB1, the data were analyzed using box plots. The graph in Figure 6 presents a comparison of descriptor *L*_Aeq,1h_ obtained by prediction methods and measurements. The statistical set obtained by the Schall 03 method has a similar variance as the statistical set obtained by measurement. The statistical file obtained using the MPVHD method has a smaller variance. The horizontal match of the upper and lower quartiles is only partial in the case of the Schall 03 method if it is almost complete in the case of the MPVHD method (width is smaller in the case of MPVHD). The median value in the case of MPVHD is closer to the value obtained by measurement than in the case of the second prediction method. From these results, the MPVHD prediction is slightly more accurate.

The second measurement in Banská Bystrica (BB2) was carried out on 14 June 2016. The meteorological conditions were again suitable for the measurement. More detailed information on the composition, number, and types of trains is in Table 3.

The results of determining quantities obtained by measurements and predictions were compared graphically (Figure 7) also in the measurement performed on 14 June 2016 in Banská Bystrica (BB2).

The maximum noise exposure was reached in the time interval from 12:00 a.m. to 1:00 p.m., although during this measurement hour only three trains passed near the BB2 location. However, one of the trains was freight, had 21 wagons, and passed in the direction of Vrútky-Banská Bystrica. The passage of the longest freight train was not recorded until the following hour. This train had 3 locomotives and 26 wagons, but it passed in the direction of Banská Bystrica—Červená Skala, i.e., on the line, which was twice as far from the measuring point. Although he passed 32 s longer, the *L*_AE_ sound pressure level A was more than 5 dB lower. Between 9:00 a.m. and 10:00 p.m., only the passage of one motor passenger train with two wagons was recorded. A motor passenger train is a special train that is relatively short and moved with non-uniform movement. It was therefore problematic to determine its speed and had a significantly different noise level than the basic noise emission level determined by both prediction models. As a result, the noise exposure at the given measuring point was the smallest. Train speeds ranged from 35 km/h to 75 km/h (average speed was 50 km/h). The equivalent sound pressure level A at the reference time “day” for BB2 was *L*_Aeq,12h_ = 61.58 dB.

The course of determining quantities obtained by measurements and prediction methods has a similar tendency in this case as well. In the time interval between 9:00 a.m. and 10:00 a.m., the predictions are significantly overestimated by both methods (especially in the case of MPVHD), at that time only one motor passenger train crossed through the assessed section. The sensitivity of both methods to this extreme situation is lower again.

Box plots were also used in this measurement. The graph in Figure 8 again presents a comparison of *L*_Aeq,1h_ levels obtained by prediction methods and measurements. The statistical dataset obtained using the Schall 03 method has a similar (albeit slightly smaller) variance to the statistical file obtained by measurement. In the case of the MPVHD method, the variance of the data is again smaller than in other cases. All variances are markedly asymmetric. The horizontal march of the upper and lower quartiles is almost complete in the case of the Schall 03 method if it is only partial in the case of the MPVHD. The median value in the case of Schall 03 is closer to the value obtained by measurement than in the case of the second prediction method. Based on these results, we can conclude that the Schall 03 method better predicts the noise situation in this measurement.

## 4. Discussion

### 4.1. Discussion of Results

All performed measurements and predictions were statistically tested. Before using the ANOVA test (analysis of variance), it is necessary to answer the question of whether the variability of the variable in the assessed sets is the same or is distinguishable. The null hypothesis H0 assumes that it does not differ. The variances between Schall 03 measurement and prediction as well as between MPVHD measurement and prediction were tested. This hypothesis needs to be confirmed—that is, the value of the *F*—the result of the Leven test and the value of the *p*—significance, which must be greater than the significance level *α* = 0.05, is interesting. In the case of differences between measured and calculated values by both prediction methods, the value of *p* is greater than the significance level *α*. For Schall 03, *p* = 0.12 and for MPVHD, *p* = 0.59. Since null hypotheses were confirmed in both cases, it is possible to use the ANOVA test (one-dimensional, multifactor).

Even in this case, the null hypothesis will be tested at the significance level *α* = 0.05: “The measured and predicted values do not depend on the location or time of the measurement or their mutual interaction.” Each factor will be tested independently for both prediction methods concerning the measurement. The test results for both methods are in Table 4 and Table 5.

The results show that the Schall 03 method is sensitive to the measurement location (the value reaches half of the significance level) and to the location–period interaction too. This is because several values were significantly underestimated in the measurement of BB1. The MPVHD method is sensitive to systematic error (absolute term) and, such as Schall 03, to the location–period interaction. This method systematically overestimates the results. A graphical interpretation of the mentioned facts is shown in Figure 9.

In the case of the prediction method Schall 03, it can be seen that the difference between the determinants obtained by the measurement and the prediction is greater in the case of the first measurements (ZV1 and BB1). Besides, the prediction at BB1 systematically underestimates the results by approximately 0.7 dB. For this reason, the second measurements (ZV2, BB2) appear to be more accurate. The results of ZV2 measurements were almost identical to the prediction and even 95% confidence intervals were the smallest.

In the case of the MPVHD prediction method (Figure 10), it can be seen that the prediction almost systematically overestimates the results. In the case of ZV1 measurement, it is approximately 2.8 dB. Only the BB1 measurement was almost indistinguishable from the prediction. The second measurements (ZV2, BB2) also appear to be more accurate with this method. Overall, however, the difference between measurements and predictions is smaller when using Schall 03, and thus this comparison is based on our comparisons as more accurate. The prediction of Schall 03 in BB1 systematically underestimates the results. The second measurements (ZV2, BB2) appear to be more accurate. The results of ZV2 measurements were almost identical to the prediction and even 95% confidence intervals were relatively small.

Table 6 compares the measurement and prediction results at both selected measurement locations in both periods. The data in dB shows how much the given method overestimates (plus sign) or underestimates (minus sign) the results of in situ measurements. It can also be seen from these values that the prediction method Schall 03 gave very similar results as the measurement, especially for ZV2 and BB2. The only significant deviation occurred at BB1, but this value is also smaller than the expanded measurement uncertainty. The MPVHD method provides slightly larger differences between the measured and predicted values, while in three of four cases it systematically overestimated *L*_Aeq,1h_. Nevertheless, these differences are also smaller than the expanded measurement uncertainty in all cases. The underestimation of the results in the case of BB2 by both predictions can be explained by the reduced smoothness of the movement of some trains in the afternoon, which these models failed to capture. Both methods increase their degree of discrepancy in extreme situations, such as with a small number of crosses per unit of time (in our case an hour), or with significantly noisier, respectively quieter trains than would correspond to the type of train and its speed. Another problem with using these methods is the poor technical condition of some trainsets.

The Schall 03 method is elaborated in more detail than MPVHD. It considers many more input parameters. The MPVHD method only describes braking with a constant (+1.6 dB), while Schall 03 deals with the proportion of disc brakes in a trainset. Only the number of wagons of a trainset enters the MPVHD method, while in Schall 03 its length is considered. In MPVHD we do not distinguish between the type of track, on the other hand in Schall 03 we do not distinguish between electric and engine traction. In the case of the Schall 03 method, the problem is the small number of train crossings if the *L*_Aeq,1h_ is less than the basic emission level of 51 dB. This applies to a train 100 m long, with a 100%-disc brake ratio and a speed of 100 km/h. In the measurement of BB2 between 9 and 10, this fact was significantly demonstrated. For similar reasons, MPVHD could not correctly evaluate this extreme.

### 4.2. Comparison of Results with Another Research

Only a few works were performed under comparable conditions by comparing measured values of environmental noise from railway transport and prediction methods. The study from Serbia [37] addressed the implementation of the Schall 03 method in the conditions of Serbian national railways. The results showed that the calculated equivalent sound pressure level A was higher than measured, with the difference in calculated values not exceeding 1.1 dB (reference time “day”). However, all the inaccuracies and insufficient possibility of adapting the method to specific conditions were much smaller than the expanded measurement uncertainty. The technological specifics of the fleet train can significantly affect the accuracy of the prediction method.

Džambas et al. studied the interim railway noise modeling method RMR was developed to create strategic noise maps for all major railways in the European Union. Traffic noise modeling in the vicinity located in Croatia’s capital Zagreb is described in that paper. Noise levels were determined using the RMR interim method, recommended by Directive, and Schall 03. Research results have shown that the noise levels determined by the RMR interim method are lower than those obtained by field measurements and that the Schall 03 noise modeling method presents more accurate results, which points to the pressing need to develop a national railway noise modeling method in Croatia. The largest deviation between the calculated and measured noise levels amounts to 3.7 dB(A) [38].

Cheng et al. evaluated the influence of axial symmetry and rotation of the wheelset on rails and found that if these parameters are not sufficiently considered in the calculations, noise can be underestimated by up to 3 dB [39]. The authors of studies from South Korea [40,41] also used the Schall 03 methodology to predict rail noise. Depending on the location of the measuring equipment, the difference between the calculated and measured values was approximately 4 dB for different types of trains. In the case of high-speed trains, the value of the difference was approximately 7 dB. It was thus confirmed that this method did not sufficiently consider the specificities of the fleet train in Korea.

In the existing work, the UK prediction method (CRN) is often used to predict railway noise, which is often used to model railway noise inside buildings. In most works, this method also overestimated the measured noise values by approximately 2 to 2.5 dB [42,43].

Pronello et al. identifies variables that have a significant effect on sound pressure levels, defines a standard procedure for noise measurement, and develop a database for setting and calibrating rail noise models. The results indicated that some of the variables of the prediction models under certain conditions can be neglected (e.g., when surrounding environmental conditions are constant, different types of trains do not cause a significant variation in noise level) [44].

Čurović et al. present a framework to manage railway noise exposure in Brazil based on a case study carried out in the city with the longest stretches of railway tracks in urban areas. Railway noise using the SRM II and ISO 9613 calculation methods maintained the aforementioned accuracy, while the current best fit CNOSSOS-EU (Railway and Industry) configuration for the study area overestimated it [45].

However, our results showed greater accuracy of both prediction models compared to the measured noise values than the results of the authors in other countries and conditions. This suggests that the Schall 03 method is suitable for the conditions of the Slovak train fleet.

## 5. Conclusions

The Schall 03 method was developed for the conditions of Germany, therefore its adoption by other countries requires the creation of additional corrections for the conditions of the train fleet, type, and quality of railway lines. It turns out, therefore, that the EU’s efforts to unify the approach in all areas of life should not involve noise prediction, as individual prediction methods have been developed in country-specific conditions and can only work in another country with appropriate adjustment settings [10].

The main aim of the work was based on the performed measurements and selected noise prediction methods to evaluate their accuracy and assess their sensitivity to systematic error (absolute term), measurement location, and time. The inaccuracy of both methods for all time intervals is less than the expanded measurement uncertainty. The result shows that Schall 03 is sensitive to the measurement site, while MPVHD systematically overestimates the results.

Measurement results of the determining quantities and their comparison with prediction methods confirmed that the prediction methods describe the actual course of measured values with sufficient accuracy. The results of the work show that the Schall 03 method is more suitable for the conditions and technological parameters of the vehicle fleet of the given locality of Slovakia than the MPVHD method. This is due to its more detailed elaboration and the national corrections too.

Testing of the prediction method Schall 03, especially in the conditions of states that have a higher technological level of railway transport, showed slightly different results compared to our work. However, they proved the sufficient suitability of this method also in conditions with higher intensity of railway traffic and in conditions of high-speed train traffic too. In the future, the post-pandemic recovery plan envisages significant investments in railway infrastructure in all countries of the European Union [9]. Noise and its effects in synergy with other factors can have a major negative impact on the physical and mental health of the human population. At present, very little or no attention is paid to these facts [46,47,48]. Prediction methods will thus be an important tool in assessing the implementation of these innovative strategies. This will make it possible to partially reduce the negative effects of environmental noise from rail transport in urban areas, especially from the point of view of human health and the psyche.

## Figures and Tables

**Figure 1 ijerph-18-07086-f001:**
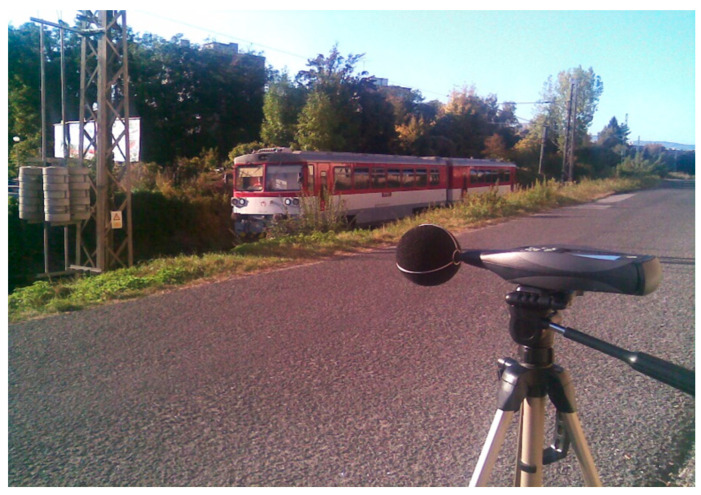
Measurement location “ZV”.

**Figure 2 ijerph-18-07086-f002:**
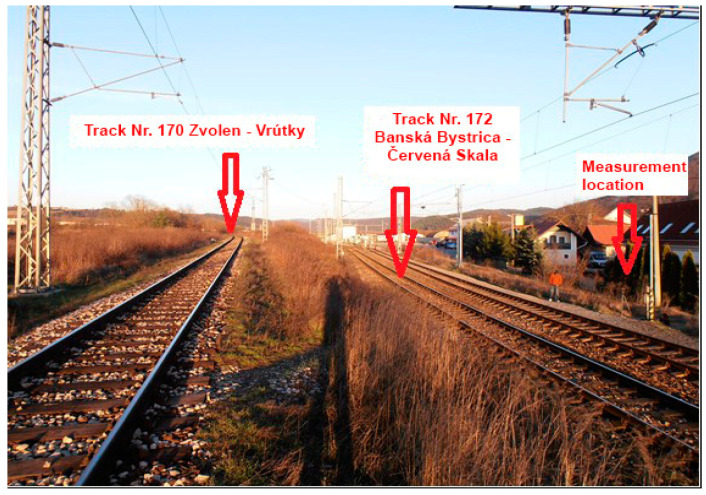
Measurement location “BB”.

**Figure 3 ijerph-18-07086-f003:**
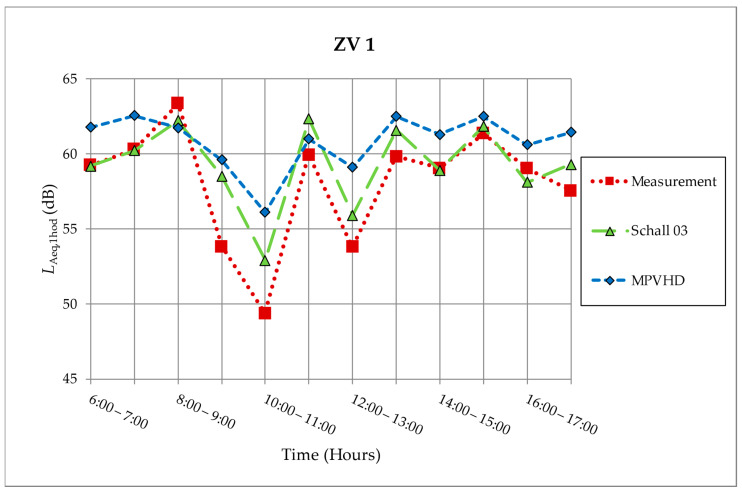
Measurement and prediction of noise from the railway on measurement location ZV1 [10,30,31].

**Figure 4 ijerph-18-07086-f004:**
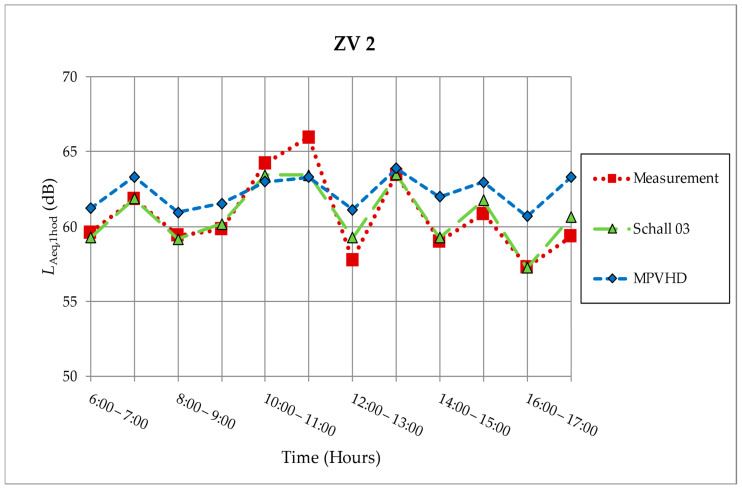
Measurement and prediction of noise from the railway on measurement location ZV2 [10,30,31].

**Figure 5 ijerph-18-07086-f005:**
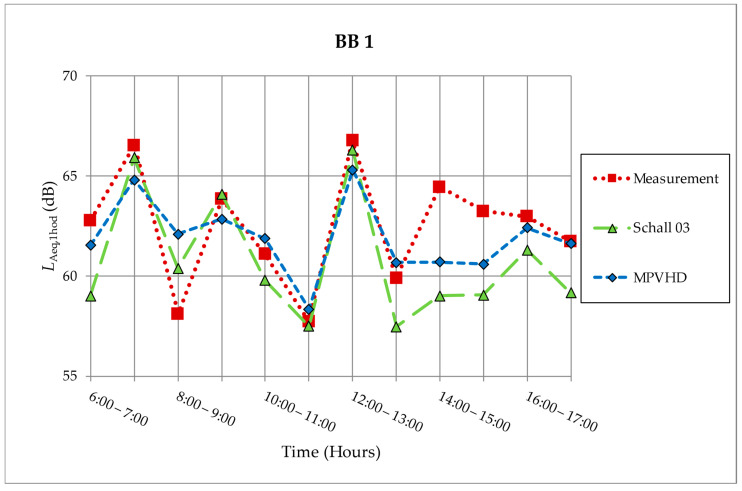
Measurement and prediction of noise from the railway on measurement location BB1 [30].

**Figure 6 ijerph-18-07086-f006:**
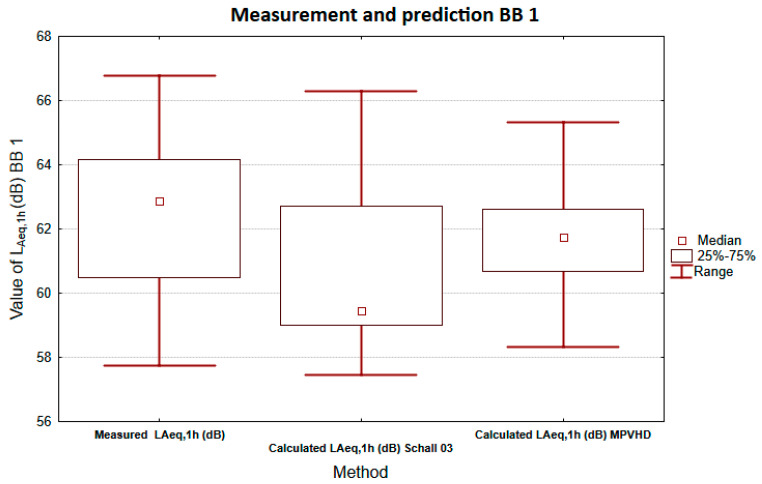
Comparison of equivalent sound pressure levels A using box graphs BB1.

**Figure 7 ijerph-18-07086-f007:**
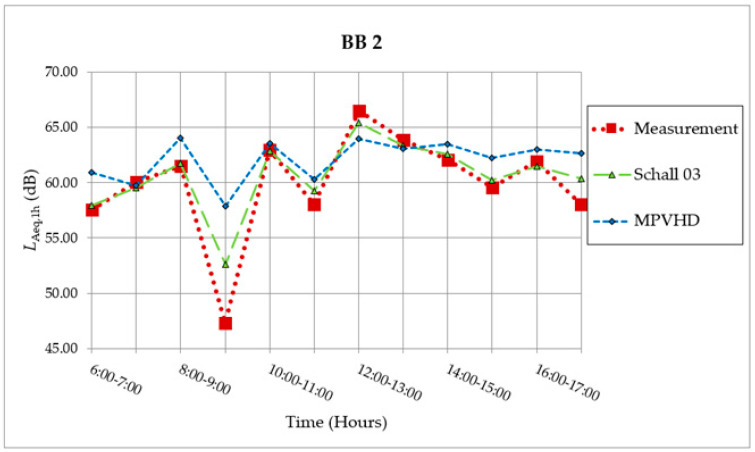
Measurement and prediction of noise from the railway on measurement location BB2 [30].

**Figure 8 ijerph-18-07086-f008:**
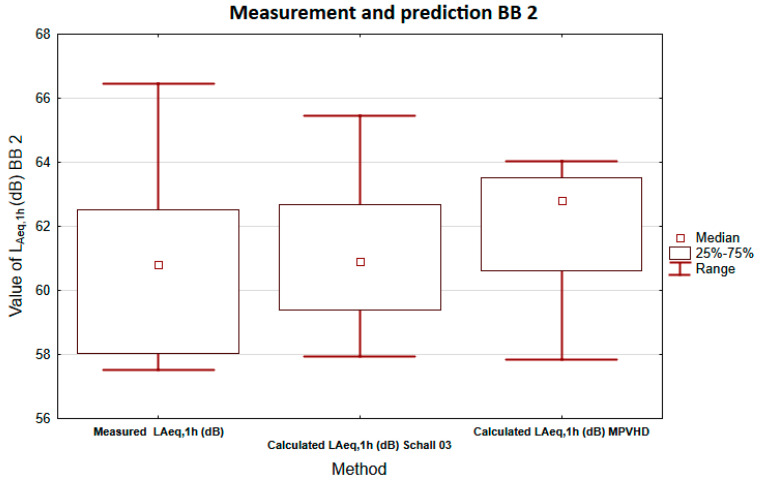
Comparison of equivalent sound pressure levels A using box graphs BB2.

**Figure 9 ijerph-18-07086-f009:**
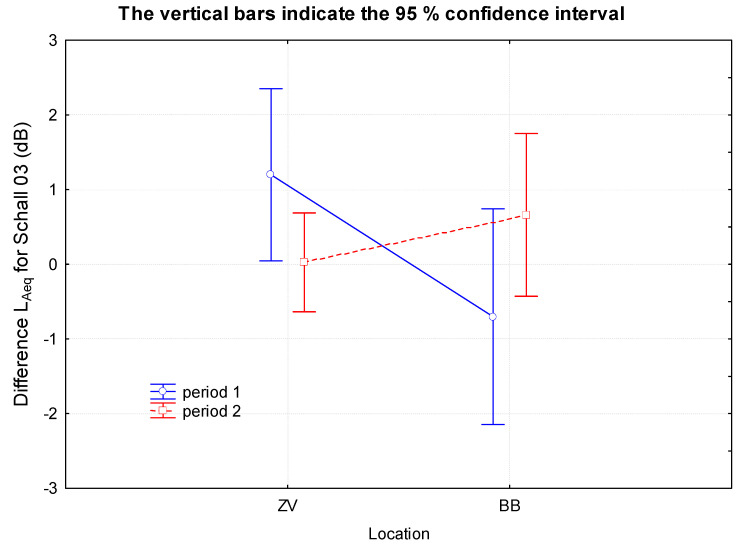
Graphical interpretation of the differences of the determining quantities obtained by Schall 03 and the measurement depending on the location and time of the measurement.

**Figure 10 ijerph-18-07086-f010:**
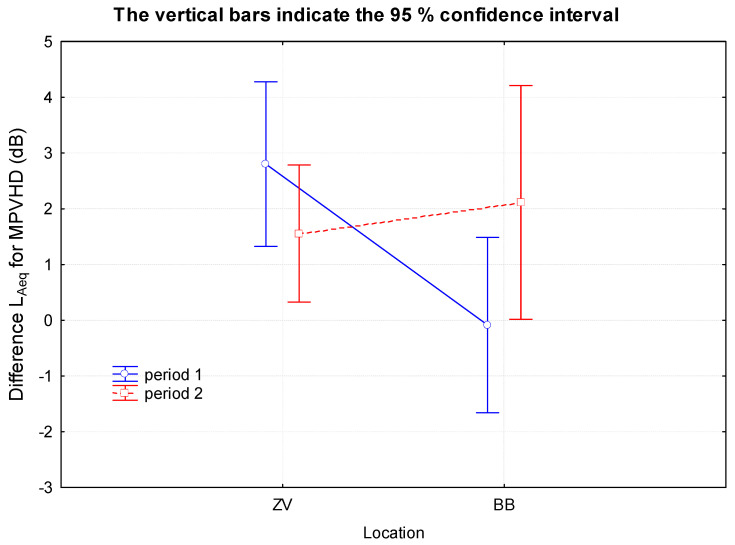
Graphical interpretation of the differences of the determining quantities obtained by MPVHD and the measurement depending on the location and time of the measurement.

**Table 1 ijerph-18-07086-t001:** Comparison of both prediction methods.

Correction/Method	Schall 03	MPVHD
Country	Germany	Czech Republic
Reference point (distance)	25 m	7.5 m
Basic relationship	$L=10\log\left[\sum^{\;}10^{\left(0.1\cdot \left(51+D_{Fz}+D_{D}+D_{L}+D_{S}\right)\right)}\right]$ $+D_{Tt}+D_{Br}+D_{Lc}+D_{Ra}$	Y=10⋅logX+40 $X=140\cdot F_{4}\cdot F_{5}\cdot F_{6}\cdot \;m$
Deceleration	$D_{d}=10\cdot \log(5-0.04\cdot P)$	+1.6 dB
Length	$D_{L}=20\cdot \log(0.01\cdot l)$	$F_{6}=0.0375\cdot z+0.5$
Speed	$D_{s}=20\cdot \log(0.01\cdot V)$	$F_{5}=0.241\cdot e^{\left(0.024v\right)}$
Type of track	Yes	No
Traction	No	engine *F*_4_ = 1.0 electric *F*_4_ = 0.65

**Table 2 ijerph-18-07086-t002:** Number and type of trains during the measurement BB1.

Direction Banská Bystrica-Červená Skala		Direction Červená Skala-Banská Bystrica		Direction Banská Bystrica-Vrútky		Direction Vrútky-Banská Bystrica	
Type	Number	Type	Number	Type	Number	Type	Number
Os	9	Os	10	Os	0	Os	1
Zr	0	Zr	0	Zr	6	Zr	6
R	1	R	0	R	0	R	0
N	2	N	1	N	1	N	1
Vú	1	Vú	1	Vú	0	Vú	0
Total	13	Total	12	Total	7	Total	8

(Os—ordinary passenger train, Zr—semi-fast regional train, R—ordinary fast train, N—goods train, Vú—maintenance train).

**Table 3 ijerph-18-07086-t003:** Number and type of trains during the measurement BB2.

Direction Banská Bystrica-Červená Skala		Direction Červená Skala-Banská Bystrica		Direction Banská Bystrica-Vrútky		Direction Vrútky-Banská Bystrica	
Type	Number	Type	Number	Type	Number	Type	Number
Os	11	Os	9	Os	7	Os	7
Zr	0	Zr	0	Zr	0	Zr	0
R	0	R	0	R	0	R	0
N	1	N	1	N	1	N	1
Vú	0	Vú	0	Vú	0	Vú	1
Total	12	Total	10	Total	8	Total	9

(Os—ordinary passenger train, Zr—semi-fast regional train, R—ordinary fast train, N—goods train, Vú—maintenance train).

**Table 4 ijerph-18-07086-t004:** One-dimensional test ANOVA for Schall 03.

Efect	Sum of Squares	Degree of Freedom	Mean Squares	*F*-Test	*p*-Level
Abs. term	0.11	1	0.11	0.03	0.846
location	**15.25**	**1**	**15.25**	**5.12**	**0.029**
period	4.16	1	4.16	1.40	0.243
location*period	**37.27**	**1**	**37.27**	**12.51**	**0.001**
Error	131.11	44	2.97		

* Statistical interaction between the location and period.

**Table 5 ijerph-18-07086-t005:** One-dimensional test ANOVA for MPVHD.

Efect	Sum of Squares	Degree of Freedom	Mean Squares	*F*-Test	*p*-Level
Abs. term	**105.70**	**1**	**105.70**	**17.65**	**0.000**
Location	23.14	1	23.14	3.86	0.056
Period	5.915	1	5.91	0.99	0.326
location*period	**45.49**	**1**	**45.49**	**7.60**	**0.008**
Error	263.49	44	5.98		

* Statistical interaction between the location and period.

**Table 6 ijerph-18-07086-t006:** Measurement and predictions–comparison.

Location	MPVHD	Schall 03
ZV1 *	+1.3 dB	+0.6 dB
ZV2 *	+0.9 dB	−0.3 dB
BB1	−1.0 dB	−1.4 dB
BB2	+0.8 dB	−0.1 dB

* some of the results have already been published in work [10].

## Data Availability

Not applicable.

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
