# Peer review of "Selected Approaches to the Assessment of Environmental Noise from Railways in Urban Areas"

_ijerph, 2021, doi:10.3390/ijerph18137086_

Round 1
Reviewer 1 Report
The manuscript "Selected approaches to the assessment of environmental noise from railways in urban areas" presented the noise assessment of railway tracks using two well-established methods. Initial result shows the overestimation of noise levels in case of MPVHD method. Whereas, Schall 03 method is more comparable to the measured noise levels. Also, the statistical test shows the location of measurement and time period are the most significant factors. The work is quite interesting, but number of measurements are not sufficient to make a general comment on the methodology. If possible, author might add few more measurements for a better statistical fit.
> Write objective of study in a clear manner in the last paragraph of introduction section.
> Why the height of the microphone was fixed at 1.5 m? Does height of the measurement affect the assessment?
> Mention the first author's name (e.g., XXX et al. [10]) in line 121.
Reviewer 2 Report
Improve the quality of the graphics.
Expand the bibliography, increase the number of references.
Authors should discuss the limitations of the Schall 03.
Specify the MPVHD method how it works, what are the basic principles, what are the limits with respect to Schall 03.
With only two measuring points how do you test the MPVHD method?
Specify the type of trains measured, speed in motion, number of wagons. If the type of train changes what happens?
What happens if the measuring point moves away from the source?
The paper does not qualify to be accepted in the journal
Reviewer 3 Report
The Schall 03 and MPVHD methods were applied to railway noise prediction in Slovakia. The results were compared with the measurements results. The first was shown to be more suitable for Slovakia.
The main problem is that the results were partially published:
- the measurements and predictions from Zvolen (ZV1) were published in (not cited!):
Nemec M. et. al., Train Noise - Comparison of Prediction Methods, Acta Physica Polonica A, Vol. 127 (2015), pp. 125-127, (especially plot from Fig. 3)
- the measurements and predictions from Zvolen: (ZV1) and (ZV2) were published in [10] (too large part of the analysis, discussion and the plots from Fig. 3).
Too large parts of text are copied from the previous papers.
A new contribution are measurements from Banska Bistrica and ANOVA tests of all data.
The paper should be reorganised in the following way:
- a thorough analysis and discussion of BB1 and BB2 data should be performed (similarly to this shown in [10]),
- the results from Zvolen should be only quoted, and taken into prediction methods assesments.
There is no reference for MPVHD method. Both Schall 03 and MPVHD methods should be introduced in a few sentences to give some background for the reader, especially variables of the prediction models should be introduced, as they are addressed in the paper.
line 278 "This is due to its more detailed elaboration and to the mentioned corrections too." - These corrections are poorly described in [10] without any reference. They should be thoroughly described also in this paper.
The analysis of the results is also poor:
line 135: Why? Both in this paper and in [10] there is no answer why SCHAll 03 method gives such result.
"However, the LAeq,1h values are different in the corresponding time intervals (the largest difference was between 10 am and 11 am, when only one train passed, which the prediction methods could not take into account correctly [10]."
line 159. The authors do not try to find the reason why the method fails.
"In the time interval between 9:00 a.m. and 10:00 a.m., the predictions are significantly overestimated by both methods (especially in the case of MPVHD), at that time only one motor passenger train crossed through the assessed section. The sensitivity of both methods to this extreme situation is lower again."
The aim stated in line 67 should be addressed further, the "various aspects" should be enumerated and thoroughly commented on.
"The aim of the work is based on the performed measurements and selected noise predictions to evaluate the accuracy of the prediction models and assess their sensitivity to various aspects (measurement location, time, etc.)."
The paper is badly organised.
Generally, the paper should be organised as follows:
- problem formulation, the aim statement (necessity for noise measurements and prediction),
- literature research (description of the methods used in the world, their characteristics (good fit to a specific state conditions, worse to another country)),
- description of chosen methods with a special emphasis on the modification or adjustments applied,
- description of the experiments (measurements) performed, with results presentation,
- analysis of the results (statistical analysis, justification of the discrepancies),
- discussion of the results (not literature, but current results, conlusions),
- summing up, the research aim should be precisely commented on (research summary, conclusions, next steps).
The paper is messy, some parts are missing (description of the methods and applied modifications, an attempt to justify discrepancies between measurements and predictions), some parts are repeated after the previous publications (with sentences taken out of the context), some parts are in an inappropriate place (in part 4. Discussion the literature examples are discussed, however it should contain discussion of this paper results).
It seems that the following reference should be taken into account:
Tamara Džambas, Stjepan Lakušić, Vesna Dragčević, Traffic noise analysis in railway station zones, Applied Acoustics, Volume 137, August 2018, Pages 27-32
The following sentences should be reformulated or clarified:
- 258 "some of the variables of the prediction models under certain conditions",
- 79 "Measurements and predictions at both measuring points were carried out repeatedly, because since 17 November 2014, based on the decision of the Government of the Slovak Republic, the Railways of the Slovak Republic have been carrying out free transport of selected population groups (students, pensioners)." (unclear taken out of the contexts from [10]).
The plots in Figs 3-4 should be corrected: right plots L axis should be scaled with less accuracy (70 instead 70,00), left and right plots should be of the same size.
Plots in Figs 5-6 may be smaller.
Round 2
Reviewer 1 Report
THE REVISED MANUSCRIPT IS MODIFIED AS PER OUR SUGGESTION.
Author Response
Thank you very much for the rewiew that helped improve our article.
Reviewer 3 Report
In my opinion some newly written parts need slight editing of English, e.g. the following sentences are unclear due to incorrect English:
142: DBr correction is considered bridges and their effect on noise (+ 3 dB).
- it should be corrected (e.g. DBr correction consideres bridges and their effect on noise)
73
The main aim of the work is based on the performed measurements and selected noise prediction methods to evaluate their accuracy and assess their sensitivity to systematic error (absolute term), measurement location, and time
75
The work also contains original results and their analysis of measurements and predictions from Banská Bystrica
***
386
The title "Conclusions" left on the bottom of the page.
Author Response
Dear rewiewer.
Thank you very much for the recommendations that helped improve the quality of our paper.
The English language was edited according to the recommendations.
The aim has been reworded:
"The aim of the work was to compare of the results measurements of sound pressure level A from the railway transport with the results of the selected prediction methods and evaluate their sensitivity to systematic error, the place and the time of measurements".
The title "Conclusions" has been moved to a new page.